# Distinct and Dynamic Changes in the Temporal Profiles of Neurotransmitters in *Drosophila melanogaster* Brain following Volatilized Cocaine or Methamphetamine Administrations

**DOI:** 10.3390/ph16101489

**Published:** 2023-10-19

**Authors:** Ana Filošević Vujnović, Lara Saftić Martinović, Marta Medija, Rozi Andretić Waldowski

**Affiliations:** 1Department of Biotechnology, University of Rijeka, Radmile Matejčić 2, 51000 Rijeka, Croatia; lara.saftic.martinovic@medri.uniri.hr (L.S.M.); marta.medija@student.uniri.hr (M.M.); randretic@biotech.uniri.hr (R.A.W.); 2Faculty of Medicine, University of Rijeka, Braće Branchetta 20, 51000 Rijeka, Croatia

**Keywords:** *Drosophila melanogaster*, cocaine, methamphetamine, neurotransmitter concentration, locomotor sensitization, LC-MS/MS

## Abstract

Due to similarities in genetics, cellular response, and behavior, *Drosophila* is used as a model organism in addiction research. A well-described behavioral response examined in flies is the induced increase in locomotor activity after a single dose of volatilized cocaine (vCOC) and volatilized methamphetamine (vMETH), the sensitivity, and the escalation of the locomotor response after the repeated dose, the locomotor sensitization. However, knowledge about how vCOC and vMETH affect different neurotransmitter systems over time is scarce. We used LC-MS/MS to systematically examine changes in the concentration of neurotransmitters, metabolites and non-metabolized COC and METH in the whole head homogenates of male flies one to seven hours after single and double vCOC or vMETH administrations. vMETH leads to complex changes in the levels of examined substances over time, while vCOC strongly and briefly increases concentrations of dopamine, tyramine and octopamine followed by a delayed degradation into *N*-acetyl dopamine and *N*-acetyl tyramine. The first exposure to psychostimulants leads to significant and dynamic changes in the concentrations relative to the second administration when they are more stable over several hours. Further investigations are needed to understand neurochemical and molecular changes post-psychostimulant administration.

## 1. Introduction

The administration of psychostimulants leads to changes in the concentration, the localization, and the chemical form of neurotransmitters crucial for their behavioral effects [1,2]. By binding to the transporters on presynaptic neurons responsible for removing excess neurotransmitters from the synaptic cleft, cocaine (COC) inhibits the reuptake and increases the concentration of dopamine, norepinephrine, and serotonin in the synaptic cleft [3,4]. Methamphetamine (METH) leads to similar changes, but by a different molecular mechanism. METH is similar in structure to dopamine and enters the presynaptic neuron through the dopaminergic transporters, as well as directly diffusing through the neuronal membrane. Once inside the presynaptic neuron, METH depletes storage vesicles of dopamine, releasing them into the cytoplasm, and eventually into the synaptic cleft by reversing the function of dopamine transporters [5]. The molecular events that follow the administration of psychostimulants are extensively studied [6,7,8], although their complexity and extensive anatomical modulation limits our understanding of the precise mechanisms that lead to the development of addiction. In *Drosophila*, the ease of genetic manipulations provides a means for describing changes in the neuronal functioning that are induced by psychostimulants [9,10]. There have, however, been no studies that address temporal dynamics in neurotransmitter concentrations after psychostimulant administration [11], preventing a comprehensive study of the role that neurotransmitter regulation has in the development of substance abuse.

*Drosophila* uses neurotransmitters such as dopamine (DA), serotonin (5-HT), glutamate (GLU), acetylcholine (ACh) and γ-aminobutyric acid (GABA), regulated by plasma membrane and vesicular transporters that are homologous to mammals [12]. In *Drosophila*, tyramine (TA) and octopamine (OA) have a homologous function to epinephrine and norepinephrine in the brain of mammals. TA and OA are present in mammals only as trace amines [13]. Neurotransmitter metabolism in flies has not been documented as thoroughly as that in mammals, but the presence of the metabolites *N*-acetyl dopamine and *N*-acetyl tyramine has been shown and measured [14]. Furthermore, the enzymes responsible for the metabolism of DA, TA and OA have been described [15]. In *Drosophila*, arylalkylamine *N*-acetyltransferase (AANAT) and arylalkylamine *N*-acyltransferase-like 2 (AANATL2) are major enzymes involved in *N*-acylation and inactivation of biogenic amines. The two enzymes have similar affinity for the acylation of DA, TA and OA.

Based on similarities of the genetic regulation and behavioral responses, *Drosophila* has been extensively used in the study of substance abuse and treatment [11,16,17,18]. The easily inducible and quantifiable endophenotype is sensitivity (SENS) to the motor activating effects of psychostimulants [19,20]. This dose-dependent behavioral response to COC and METH can be induced by oral administration, or exposure to volatilized psychostimulants, and has been well studied and used to screen for new candidate genes related to substance abuse [21,22,23]. In *Drosophila*, locomotor sensitization (LS), or progressive and persistent augmentation of behavioral or locomotor response to the repeated dose of the same concentration, has been described only after volatilized exposure [20,21,24]. In rodents, sensitization has been shown to correlate with the enhanced predisposition to self-administer psychostimulants [25]. Although the existence of sensitization in humans is debated, it has been proposed to correspond with traits, such as compulsive drug-seeking behavior [26,27]. Most importantly, LS represents an expression of the changes in the neuronal homeostasis that involves transcriptional and epigenetic changes [28], which are the basis of the neuronal plasticity induced by substance use [29]. Since the induction (SENS), and the expression (LS) of neuronal plasticity are initiated by dynamic changes in neurotransmitter release following psychostimulant administration, description of those changes could aid in the understanding of the downstream molecular changes that are linked to the development of addiction-related behaviors. 

Currently, there are no published reports about temporal changes in neurotransmitter concentration following administration of vCOC or vMETH in *Drosophila*. Since these two psychostimulants have a distinct mechanism of action and initiate a cascade of molecular and biochemical events, it seems reasonable that this leads to a different minimal time interval needed to induce the development of LS to repeated administrations of vCOC and vMETH [20,24]. A description of the changes in the released neurotransmitters following vCOC or vMETH administration should be useful for a better understanding of the pharmacokinetics and pharmacodynamics that are the basis of the similarities and differences between behavioral consequences of vCOC and vMETH administration.

To this end we examined the temporal profile of changes that occur at the level of the whole brain. We quantified the concentration of neurotransmitters dopamine (DA), octopamine (OA), and tyramine (TA). We also semi-quantified glutamate (GLU), acetylcholine (ACh) and γ-aminobutyric acid (GABA), as well as non-metabolized cocaine (COC) and methamphetamine (METH), and DA metabolite *N*-acetyl dopamine (NaD) and TA metabolite, *N*-acetyl tyramine (NaT). We applied the LC-MS/MS analysis to whole heads homogenates following one and two administrations of vCOC or vMETH, and analyzed the extracts one, three, five and seven hours post-administration.

## 2. Results

The administration of a single dose of vCOC or vMETH to *D. melanogaster* leads to an increase in locomotor activity, or the sensitivity (SENS). Flies develop locomotor sensitization (LS) to the second exposure when it is given at least six hours later for vCOC, and ten hours later for vMETH [20,24]. This likely reflects different changes in the neurotransmitter concentrations that occur following vCOC and vMETH exposures. To investigate if and how neurotransmitters change following vCOC and vMETH administrations, we first measured the concentration of the active forms of cocaine and methamphetamine, and the monoamines linked to psychostimulant responses: DA, OA and TA, from one to seven hours after the administration of one or two doses of vCOC or vMETH.

### 2.1. Distinct Temporal Profiles of Monoamine Concentrations after vCOC and vMETH Administrations

The concentration of the non-metabolized COC gradually declined after the first administration, but a significant amount of COC remained after seven hours (Figure 1A). This likely led to the higher starting COC concentration at one hour after the second exposure (Figure 1A). The slope of the gradual decrease in the concentration of COC was similar after the first and second administration (Figure 1A). This finding was unexpected considering the metabolism of cocaine in mammals [3] and suggesting a distinct mechanism of cocaine degradation in *Drosophila*. There was a pronounced peak in DA concentration three hours after the administration of a single dose of vCOC, which returned to the pre-peak levels at five and seven hours (Figure 1B). This contrasted with DA concentration after the second dose of vCOC when the levels remained similar over the observed period. We observed similar dynamics for the concentration of tyramine (TA) and octopamine (OA) (Figure 1C,D); there was a sharp increase in the third hour after the administration, which returned to the levels observed at one, five and seven hours post-exposure. The second exposure did not induce any significant modulation, and levels overlapped with the level at one-hour post first exposure.

The concentration of the non-metabolized METH gradually declined after the first administration, but a significant amount of METH remained after seven hours (Figure 2A), similar to vCOC. This likely led to the higher starting METH concentration at one hour after the second exposure (Figure 2A). However, after the second exposure, degradation was very rapid and was not detectable after seven hours, suggesting that there are more effective mechanisms of degradation triggered after vMETH exposure relative to vCOC exposure. The effect of vMETH on DA concentration differed significantly after the first and second exposure, and relative to changes induced by vCOC. The highest concentration of DA after the first vMETH administration was present after one hour and then gradually declined and leveled off. In contrast, after the second administration, DA concentration was higher at five and seven hours and lower for the first two time points (Figure 2B). TA and OA similarly had a different concentration temporal profile compared to vCOC administration. The significant increase after the first exposure at three hours was absent, and both TA and OA had a small decrease over time (OA) (Figure 2D), or mostly level amounts (TA) (Figure 2C). Concentrations of TA and OA after the second exposure of vMETH were like the first, except for a statistically significant increase in the OA concentration at seven hours (Figure 2D).

Thus, the important difference in the temporal profiles was that vCOC leads to large peaks of DA, OA and TA at three hours after the first administration which were absent after the vMETH administration. After the second administration of vCOC, concentrations of DA, OA and TA remained stable over time, while after administration vMETH levels were less stable and tended to increase at later time points (Appendix A in Appendix A).

### 2.2. Temporal Profiles of Dopamine and Tyramine Metabolites after vCOC and vMETH Administrations

Since vCOC, unlike vMETH, administration significantly affected DA and TA, we measured the relative concentration of the DA metabolite, *N*-acetyl dopamine (NaD), and the TA metabolite, *N*-acetyl tyramine (NaT), after vCOC or vMETH exposures. NaD concentrations gradually increased after the first vCOC and vMETH administrations, reaching the highest levels at seven hours (Figure 3A,B). After the second administration NaD levels were, in general, higher than after the first (Figure 3A,B), suggesting an increase in the DA degradation and the activation of AANAT and AANATL2. After the second administration of vCOC and vMETH, NaD levels exhibited peaks, a moderate one at five hours after vCOC exposure (Figure 3A) and a more significant one three hours after vMETH exposure (Figure 3B).

NaT concentration had a distinctive peak at five hours after the first vCOC administration (Figure 3C), two hours later then the observed peak of TA (Figure 1C), indicating that the significant amount of TA was metabolized to NaT, but with a delay of two hours. The levels of TA and NaT correlated after the first vMETH administration without the time delay; the levels after vMETH administration are relatively low and stable over the entire time course (Figure 2C and Figure 3D). Correlations between the concentration of TA and NaT extended for the second administration of vCOC and vMETH. NaT levels after the second vCOC administration paralleled the levels of TA without large fluctuations (Figure 1C and Figure 3C). Similarly, there was no time delay between the TA peak and NaT at three hours after the second vMETH administration (Figure 2C and Figure 3D). 

After the second vMETH administration, the levels of NaT were, overall, significantly higher than after the first administration, a profile that was distinct from vCOC (Figure 3C,D). A direct comparison of NaD concentrations after the first vCOC and vMETH exposures shows higher values for vMETH vs. after the second exposure (Appendix A in Appendix A). NaT concentration differed between vCOC and vMETH at five hours after the first exposure, where vCOC led to a spike, and after the second exposure at three hours when there was a spike after the vMETH exposure (Appendix A in Appendix A).

### 2.3. Temporal Change after vCOC and vMETH Exposure of the Excitatory and Inhibitory Neurotransmitters

Considering that there is relatively limited focus of how glutamate (GLU) and acetylcholine (ACh) regulate *Drosophila’s* behavior in general, and the psychostimulant-induced effects in particular [30], we quantified temporal changes in GLU and ACh following vCOC and vMETH administration. We also quantified the amount of γ-aminobutyric acid (GABA), the primary inhibitory neurotransmitter in mammals as in *Drosophila*. 

Similar to the monoamines, concentrations of GLU and ACh were higher after the first administration of vCOC and vMETH than after the second (Figure 4A–D). However, unlike the monoamines, neither vCOC nor vMETH induced large spikes in the concentration of GLU and ACh. Furthermore, the significant changes occurred at later time points; for GLU, five hours after vCOC (Figure 4A), and at seven hours after vMETH administration (Figure 4B), and for ACh five and seven hours after the administration of either psychostimulant (Figure 4C,D). 

The temporal profile of GABA had two unique differences relative to all other neurotransmitters. First, it had a sharp increase at seven hours after the first vCOC administration (Figure 4E), while all other neurotransmitters showed much earlier peaks. Second, GABA had a peak at five hours after the second vMETH administration (Figure 4F), while other neurotransmitters showed little variation over time after the second vCOC or vMETH administration. Direct comparisons between vCOC- and vMETH-induced changes in the concentration of GLU, ACh and GABA are presented in Supplementary Figure (Appendix A in Appendix A).

## 3. Discussion

Neurotransmitter systems in the brain are highly complex, with specific neuronal populations exhibiting distinct functional roles in behavioral regulation. The difference in the vCOC and vMETH dosing intervals that lead to the development of LS is likely related to the difference in the pharmacokinetics of those psychostimulants in the specific brain areas. However, reports about the temporal regulation of the neurotransmitter concentration, even in the whole brain of flies, are still lacking. To systematically examine changes in the concentration of several excitatory and inhibitory neurotransmitters in the whole head homogenates of male flies, we used the LC-MS/MS method with two important metabolites and non-metabolized COC and METH across the span of one to seven hours after the administration of single and double volatilized COC (vCOC) or METH (vMETH). We show that DA, TA, and OA show sharp and brief concentration spikes after the first vCOC administration, but they are lacking after vMETH administration. The common element between vCOC and vMETH is that there is a greater variation in the concentration of monoamines after the first than after the second administration. Peak concentrations of DA and TA metabolites, NaD and NaT, show a time delay relative to peak DA and TA concentrations, but only after the first vCOC administration where the peaks are brief and pronounced. In all other experiments, the level of metabolites mostly overlaps with the amount of DA and TA. These findings provide valuable insights into the neurochemical dynamics associated with COC and METH exposure in *Drosophila* and their mechanism of action. 

After a single dose of vCOC, there is a significant increase in DA concentration, limited to three hours post-administration. This agrees with reports that show an increase in DA concentration outside the cell after COC exposure [31], and with the motor-activating effects that occur shortly after COC exposure [19,20,21]. The DA concentration after the second dose is higher at one hour compared to the first administration, and this difference could correlate with the increased motor-activating effects expressed as LS. These results also agree with reports showing that genetic or pharmacological manipulation of DA concentration influences sensitivity and LS to COC and other abused substances in *Drosophila* [20,22,32]. Considering that DA is produced by a small number of DA positive neurons grouped in distinct clusters with specific projections and functions in different behaviors, our measurements at the level of the whole brain are relatively crude. To define anatomical regions of fly brain that have a specific role in the motor-activating, arousing, and rewarding aspects of psychostimulants, future studies will have to employ more sophisticated measurements of the localized changes in DA and other monoamines. Namely, the significant increase in the locomotor activity following vCOC administration occurs within the first ten minutes post-administration, upon which it returns to the baseline, while we have detected the peak DA concentration at three hours post administration, when locomotor activity is at the baseline levels [20]. Because DA concentration is important for the motor-activating effects of psychostimulants [22,23,33], our findings suggest that there is likely to be an anatomically localized increase in DA transmission sufficient for motor-activating effects which is not detectable in the whole brain homogenate. The significant but delayed increase that we detect at the level of the whole brain is likely a consequence of the change in the transcriptional regulation that is initiated after the first administration, which eventually results in the expression of locomotor sensitization as a type of neuroplastic change. 

Flies and other insects do not appear to further process dopamine to noradrenalin or adrenalin, but rather use the structurally similar neurotransmitters TA and OA [13], both classified as trace amines in mammals [34]. We have measured a large surge in TA and OA concentrations at three hours following the first administration and relatively low levels at all other time points, including after the second administration. Our measurements agree with the known role of TA in vCOC-induced behavioral sensitization [35]. Although TA is converted to OA in one step by the enzyme tyramine beta-hydroxylase, it was shown that it is TA, not OA, that is required for the development of behavioral sensitization following repeated vCOC administrations [35]. The increase in TA concentration that we observed agrees with the role that TA has in the long-term behavioral effects induced by vCOC and the time course of the induction of the enzyme activity of tyrosine decarboxylase that converts tyrosine to TA [35]. Considering that we observed almost identical temporal and quantitative amounts of TA and OA, this would indicate that tyramine beta-hydroxylase rapidly converts TA to OA. At this point, it is not known if OA might play a role in another aspect of vCOC-induced behavior that has thus far not been explored. The initial peak in DA, OA, and TA concentrations observed three hours after the first vCOC dose does not occur after the second dose, showing a striking difference in the neurochemical response to repeated vCOC exposure and an important difference in the temporal dynamics of monoamine regulation that is associated with the development of LS. 

Although flies show similar sensitivity to the motor-activating effect of vMETH and develop LS [24], the minimal time interval for the development of LS to vCOC is six hours, while to vMETH it is ten hours. Pharmacological or genetic manipulation of DA similarly prevents the development of LS to METH [24], and METH induces a rapid increase in the synaptic DA concentration shortly after application [31]. Similarly, we have observed the highest DA levels at one hour after the first administration followed by a gradual decrease in DA concentration over seven hours. This might indicate that in flies, as in rodents, there is a decrease in the number of DA uptake sites followed by a decrease in the activity of the enzyme tyrosine hydroxylase, the rate limiting step in DA synthesis [36,37]. Furthermore, the high initial levels of DA after the first administration indicates that the motor-activating effects of METH that are present minutes after the exposure to METH, and are DA-dependent, are happening due to an increase in DA on a relatively large scale throughout the brain. However, the second vMETH administration results in an opposite profile of DA concentration, starting initially low and steadily increasing over the course of seven hours. Considering the role that DA has in the regulation of arousal [38,39,40], and other behaviors [41,42,43,44], this increase should be explored in the context of the long-term changes in brain physiology that occur as a consequence of vMETH exposure.

Relative to the surge in TA and OA following the first vCOC administration, the changes following the first vMETH administration are relatively modest. There is a small but significant increase in OA concentration at five hours after the first vMETH administration and at seven hours after the second. The observed changes in OA are interesting in the context of a potential interaction between dopaminergic and octopaminergic regulation. In mammals, trace amine associated receptor 1 (TAAR1) serves as a direct intracellular target for amphetamines in DA neurons [45]. The activation of TAAR1 by amphetamines triggers a signaling cascade that leads to the release of stored DA into the synapse [46,47]. After the first vMETH administration, TA, unlike OA, does not change much over the observed time course, but has a minor peak at three hours after the second administration. This is unique in two aspects: first, temporal profiles of TA and OA do not temporally correlate as they do after vCOC administrations, and second, the amount of TA has a distinct increase after the second but not after the first vMETH administration. This suggests important differences in the role those different monoamines, including DA and OA, play in the motor-activating effects of vCOC and vMETH. 

DA and TA are metabolized through acetylation by arylalkylamine *N*-acetyltransferase (AANAT) [48] and arylalkylamine *N*-acyltransferase-like 2 (AANATL2) [15] into *N*-acetyl dopamine (DaT) and *N*-acetyl tyramine (NaT). After the first vCOC administration, there is a gradual increase in the concentration of NaD over seven hours, with a significant increase two hour after the DA spike, suggesting that the acetylases are metabolizing DA to control the vCOC-induced DA increase. Similarly, a sharp increase in NaT occurs with a two-hour delay relative to the TA peak, indicating that most of the TA has been metabolized to NaT. However, in both cases, there is a time lag of two hours present between the DA and TA peak and the metabolite peak. There are two potential explanations for this dynamic. First, new AANAT and AANATL2 must be synthesized to metabolize the excessive amounts of DA and TA. Second, the existing amount of the enzymes requires a longer time to metabolize the substrate. We favor the first explanation, particularly in the case of the TA to NaT conversion after the first vCOC administration, because if the second explanation is correct, we should be able to detect a more gradual increase in NaT. Since in all other measurements we do not observe any time lag, we suspect that the enzymes that are present are sufficiently active to metabolize DA and TA. However, in all those instances, the levels of DA and TA are not excessively high, further supporting the potential dual mechanism of *N*-acetylases action to maintain the physiological levels of DA and TA, one that involves new synthesis and another that involves rapid turnover (Figure 5). 

In *Drosophila*, it is not known what role GLU, ACh and GABA have in the regulation of psychostimulant-induced behaviors. The temporal dynamics of GLU and ACh after the first or second administration of vCOC and vMETH are lacking abrupt or extensive changes in concentration, as we have described for the monoamines. However, considering that GLU transmission is a primary contributor in the majority of neuroplastic processes related to abused substances in mammals [49], minor, but significant changes that we recorded after the first vCOC and vMETH administration could potentially have a similar impact on the modulation of specific neuronal networks in flies that are related to addiction endophenotypes. Since both GLU and Ach show moderate increases at the later time points after the first psychostimulant administration, it could indicate that these changes are driven by pharmacodynamics of the initial vCOC and vMETH monoaminergic targets.

In mammals, the function of the inhibitory neurotransmitter GABA is intricately linked with the release of DA in the striatum where it modulates behaviors such as motivation and reinforcement learning [50] and inhibits DA release by binding to GABA receptors on the dopaminergic neurons. Since we have observed a sharp increase in GABA concentration at seven hours after the first vCOC administration, it would be interesting to investigate if that change relates to the relatively low and stable levels of DA after the second vCOC administration. Similarly, after the second administration of vMETH, an increase in GABA precedes a decrease in DA by two hours.

## 4. Materials and Methods

### 4.1. Fly Strains 

We used 3–5 days old male *D. melanogaster* wild-type (*wt*) strain *Canton S*, grown on a standard nutrient medium based on cornmeal and agar at 25 °C with 70% humidity in a 12-h light/12-h dark cycle. Using CO_2_ anesthesia and a microscope, adult males were collected and transferred to plastic vials with corn food until complete recovery, after which they were transferred to the FlyBong apparatus. The corn food contains sugar (7.9%), inactivated yeast (5.0%), agar (0.8%), corn flour (7.2%), 15 mL 10% antifungal methyl parahydroxybenzoate (≥98%, Carl Roth, Karlsruhe, Germany) dissolved in 96% ethanol (GramMol, Zagreb, Croatia) and 8 mL of propionic acid (≥99.5%, Sigma Aldrich, Buchs, Switzerland).

### 4.2. Psychostimulant Delivery Using FlyBong

The FlyBong enables the delivery of a precise dose of volatilized COC (vCOC) and volatilized METH (vMETH) to individual flies housed in the polycarbonate tubes placed in a monitor that holds 32 tubes (Drosophila Activity Monitoring System (DAMS), TriKinetics, Waltham, MA, USA) [20,24,32]. The tubes contain corn food on one side to prevent dehydration and starvation and two small holes that allow the aerosol to flow through the system. First, a volatilization chamber (three-necked flask in a heating cap) was used, which is connected on one side to the air pump, and on the other side to the dispenser attached to the DAMS monitor, using rubber tubing. The central opening of a three-necked flask is used to pipet 75 μL of a 10 mg/mL solution of cocaine hydrochloride or methamphetamine hydrochloride dissolved in 95% ethanol. The solution is added four to six hours before the experiment to allow for ethanol evaporation. The closed flask is heated to 185 °C for 8 min. During this time, the valve on the rubber tube connecting the flask and the dispenser is closed. After 8 min, the heating cap is turned off, the air pump is turned on, and the valve opens, during which time the aerosol is delivered with a flow rate of 150 L/h for 1 min to the dispenser and to the polycarbonate tubes containing individual flies. Administration of vCOC was repeated twice a day (09:00 and 15:00) and for vMETH (09:00 and 19:00) with control experiments in which the flies were exposed only to the heated air. Experiments were conducted in an incubator at 24 °C and a humidity of 75% under alternating light and dark conditions (12 h:12 h). 

### 4.3. Sampling and Sample Preparation

After drug administration using the FlyBong, the flies remained in the monitors and 15 flies were randomly collected one, three, five and seven hours after one and two vCOC and vMETH administrations, and the same for the controls exposed to heated air (Figure 6). Flies were then placed on ice, and their heads were severed using dissection scissors and tweezers. We optimized the protocol according to published methods for neurotransmitter quantification in *Drosophila* head homogenates [51,52]. Fifteen (15) heads per sample were collected and mechanically homogenized on ice in 300 μL of 0.1 M perchloric acid (95%, Sigma Aldrich, Buchs, Switzerland) for 15–20 s. Immediately after homogenization, the samples were centrifuged for 45 min at 14,000 rpm and 4 °C. The samples were then filtered through a cellulose filter with 0.20 μm pores (Ma-cherey-Nagel, Chromafil Xtra RC-20/13, 0.20 μm, 13 mm) into a 1 mL transparent vial and closed with a stopper containing a septum.

### 4.4. Liquid Chromatography with Tandem Mass Spectrometry (LC-MS/MS)

An Agilent 1260 series HPLC chromatograph equipped with a degasser, binary pump, autosampler, and column oven was coupled to an Agilent 6460 triple quadrupole mass spectrometer (QQQ) with an AJS ESI source to perform quantitative analyses of the monoamines DA, OA, and TA as well as semi-quantitative analyses of GLU, ACh, GABA, COC, METH, DA metabolite *N*-acetyl dopamine (NaD) and TA metabolite *N*-acetyl tyramine (NaT). A Purospher STAR RP-18 Hibar HR column (50 mm × 2.1 mm, 1.7 m, Merck, Darmstadt, Germany) was used for chromatographic separation. The mobile phase was composed of (A) 1% formic acid in miliQ water and (B) acetonitrile. The gradient elution was as follows: 0–0.9 min linear gradient from 1% to 10% B, 0.9–3 min from 10% to 20% B, 3–4.5 min from 20% to 25% B, 4.5–6 min from 25% to 30% B, 6–6.1 min from 30% B to 99% B, 6.1–6.2 min from 99% B to 1% B and 6.2 to 10 min 1% B. Post time was set to two min. The flow rate was 0.33 mL/min. The column oven was maintained at 25 °C. The sample injection volume was 2.5 µL. All samples were injected in triplicate. For AJS-ESI-QQQ, the parameters were set as follows: capillary voltage was 3.5 kV in both positive and negative modes, nozzle voltage was 0.5 kV, ion source temperature was set to 300 °C, gas flow was 5 L/min, nebulizer pressure was 45 psi, drying gas temperature was 250 °C, and sheath gas flow was 11 L/min. Nitrogen was used as the collision gas. The MSMS optimization parameters are listed in Appendix A (for compounds quantified with standards) and Appendix A (for compounds that were semi-quantified without standards). For compounds that were semi-quantified without standards, extracted ion chromatograms (EIC) of deprotonated ions were used for calculations. Data processing was performed using MassHunter Qualitative Analysis version B.07.00 (Agilent Technologies, Santa Clara, CA, USA).

### 4.5. Data Analysis and Statistics

All the data from MassHunter Qualitative Analysis software v. B.07.00 (Agilent Technologies, Inc., Santa Clara, CA, USA; 2014) were processed by MS Excel to calculate the concentration ratio between experimental groups exposed to the volatilized compound and the control group exposed to hot air. Statistical analyses and visualizations were performed using Prism 10.0.2 (GraphPad, La Jolla, CA, USA). Differences between treatments at different time points were analyzed using two-way ANOVA, followed by Tukey’s multiple comparisons test, depending on the data set. All data were tested for normality using Bartlett’s test or Brown–Forsythe’s test. Differences were considered significant if *p* < 0.05.

## 5. Conclusions

The major findings of this study include a depiction of the distinct and dynamic changes in the concentration of several major neurotransmitters and their metabolites in the fly brain following cocaine or methamphetamine administration. The complex changes in the profile of neurotransmitters following vMETH administration are in line with multiple cellular targets of vMETH relative to vCOC. We confirmed the importance of DA and TA in response to vCOC, and a delay in the appearance of their metabolites. Such a delay is not present in cases where neurotransmitters do not show a high increase in their concentration, such as with second exposures to vCOC or vMETH. In general, concentrations after second exposures are more stable than after the first, suggesting potential adaptations at the level of synthesis, release, and degradation of the neurotransmitters. We have shown, for the first time in *Drosophila*, temporal profiles of GLU, ACh and GABA, which all show a more moderate and delayed increase in their concentration. Although the measurements were performed on whole head homogenates, which are not representative of localized changes at the level of the specific neuronal groups that are important in the control of different behaviors, our study generated several important findings that require further work to gain a better understanding of the neurochemical and molecular changes that occur after psychostimulant administration.

## Figures and Tables

**Figure 1 pharmaceuticals-16-01489-f001:**
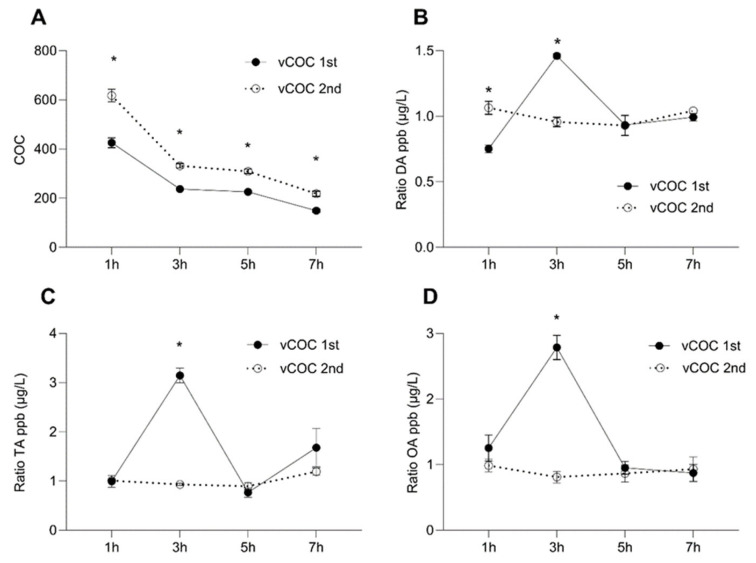
Temporal dynamics of changes in the concentration ratio of non-metabolized cocaine (COC) and dopamine (DA), tyramine (TA) and octopamine (OA) after the vCOC administrations. The concentration ratio of COC (**A**), DA (**B**), TA (**C**) and OA (**D**), was measured in the heads 1, 3, 5 and 7 h after one administration of vCOC (75 μg) at 09:00, and two administrations (2 × 75 μg) at 9:00 and 15:00. vCOC was administered using the FlyBong method. Two-way ANOVA with Tukey’s multiple comparisons test. *: *p* < 0.05.

**Figure 2 pharmaceuticals-16-01489-f002:**
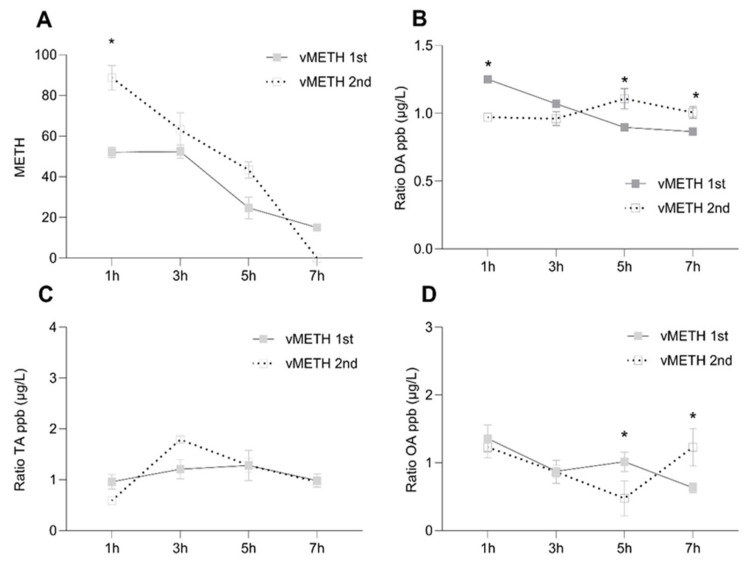
Temporal dynamics of changes in the concentration ratio of non-metabolized methamphetamine (METH) and DA, TA and OA after the vMETH administration. The concentration of METH (**A**), DA (**B**), OA (**C**) and TA (**D**), was measured in the heads 1, 3, 5 and 7 h after one administration of vMETH (75 μg) at 09:00, and two administrations (2 × 75 μg) at 9:00 and 15:00. vMETH was administered using the FlyBong method. Two-way ANOVA with Tukey’s multiple comparisons test. *: *p* < 0.05.

**Figure 3 pharmaceuticals-16-01489-f003:**
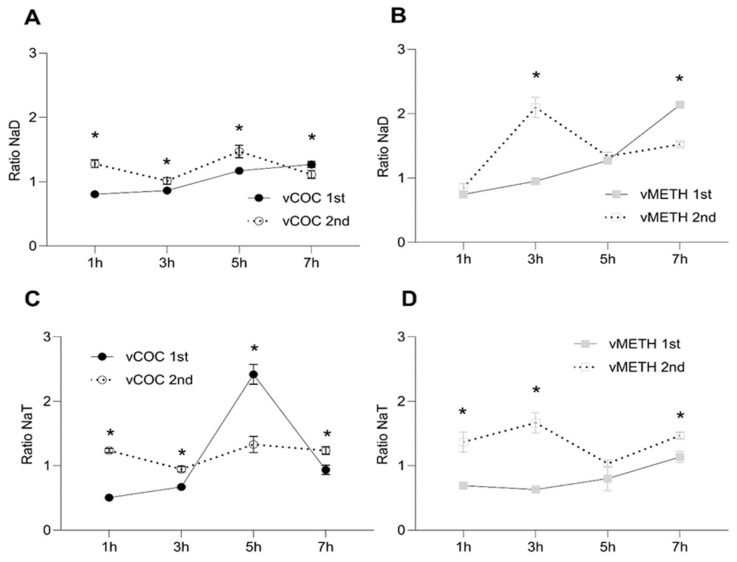
Temporal dynamics in the concentration ratio of DA metabolite *N*-acetyl dopamine (NaD) and TA metabolite *N*-acetyl tyramine (NaT) semi-quantified without standards after vCOC and vMETH administration. Changes in the concentration of NaD (**A**) and NaT (**C**) after the administration to one (vCOC first) and two (vCOC second) doses, and NaD (**B**) and NaT (**D**) after administration of one (vMETH first) and two (vMETH second) doses. Two-way ANOVA with Tukey’s multiple comparisons test. *: *p* < 0.05.

**Figure 4 pharmaceuticals-16-01489-f004:**
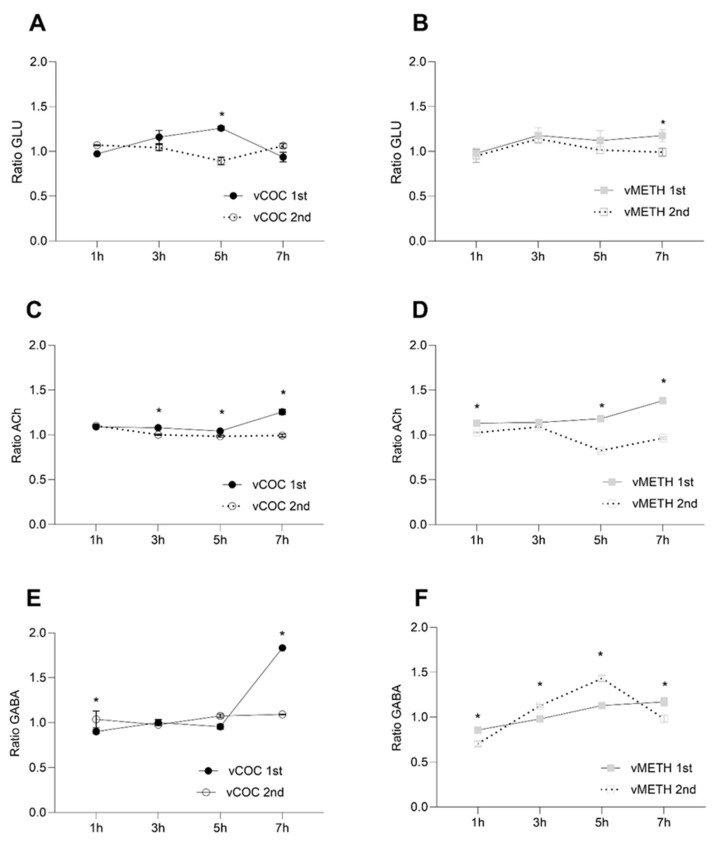
Changes in the concentration ratio of glutamate (GLU), acetycholine (ACh) and γ-aminobutyric acid (GABA) after one and two vCOC and vMETH administrations. The ratio of the relative proportion of GLU (**A**,**B**), ACh (**C**,**D**) and GABA (**E**,**F**) was measured in the extract of head homogenates at 1, 3, 5 and 7 h after one administration of vCOC or vMETH (each 75 μg) administered at 9:00, and two administrations (2 × 75 μg) at 9:00 and 15:00 for vCOC and (2 × 75 μg) at 9:00 and 19:00 for vMETH using the FlyBong method. Two-way ANOVA with Tukey’s multiple comparisons test. *: *p* < 0.05.

**Figure 5 pharmaceuticals-16-01489-f005:**
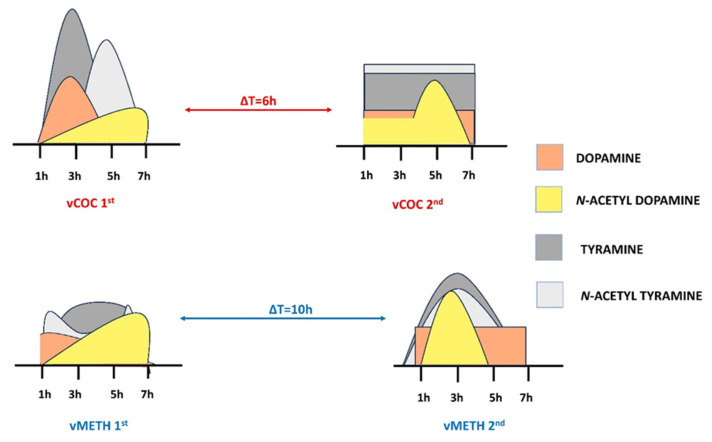
Presentation of the time course of DA and TA versus their metabolites NaD and NaT. Time lag in metabolites is present in cases of high spikes in DA and TA, while the metabolite levels correlate with the substrate levels when the concentrations are uniform.

**Figure 6 pharmaceuticals-16-01489-f006:**
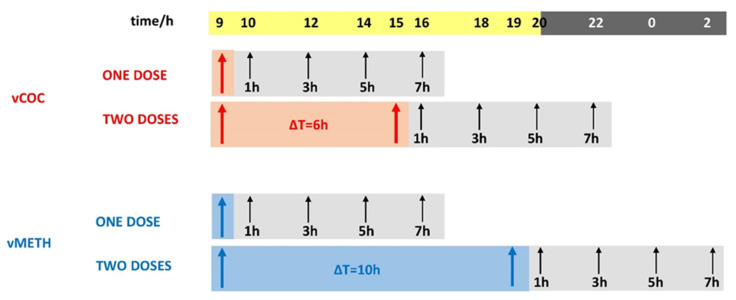
Presentation of the time course of the vCOC and vMETH administrations and time points for the sample collections and preparation of the head extracts used for the quantification of neurotransmitters.

## Data Availability

Data is contained within the article and Appendix A.

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
