# Peer review of "Distinct and Dynamic Changes in the Temporal Profiles of Neurotransmitters in Drosophila melanogaster Brain following Volatilized Cocaine or Methamphetamine Administrations"

_pharmaceuticals, 2023, doi:10.3390/ph16101489_

Round 1

Reviewer 1 Report

1.      The article lacks novelty, as several lines of articles pertaining to role of cocaine or methamphetamine in Drosophila brain including psychology and locomotor activity have been published. Even though authors have tried to put out the novelty of the study, can the authors emphasize and convince more specifically on the novelty part? Furthermore, how it would be relevant to human and clinical aspects?

2.      Why only male flies were investigated specifically? Is there any hormonal influence in these characteristics associated with addiction and locomotion?

3.      Are 1-7 hrs. of duration sufficient to produce the effects. What was the age of flies?

4.      Is method required to be mentioned in the Introduction (as mentioned at the end of this section)?

5.      How did the authors evaluate addictive and locomotor activity in flies? Aren’t rats/mice a better option for behavioral studies like addiction and locomotion than flies, which are instead good for genetic studies?

6.      Why was the method section put after the result and discussion? If this is the journal requirement the okay.

7.      Line 122, reference can be provided in the same uniform style rather than doi.

8.      What could be the reason for elevation of metabolized cocaine after second administration but not that of methamphetamine?

9.      Was there any possibility to study the specific brain region rather than taking the whole fly brain?

10.  The study lacks in terms of findings related to molecular and genetic level.

11.  Some of the very old references can be updated with the recent ones, if possible.

The English language is okay. Minor corrections can be done.

Reviewer 2 Report

Title: Distinct and dynamic changes in the temporal profiles of neurotransmitters in D. melanogaster brain following volatilized cocaine or methamphetamine administrations.

The present study elucidates alterations in the concentration of several major neurotransmitters, including dopamine, octopamine, tyramine, glutamate, acetylcholine, and γ-aminobutyric acid, as well as their respective metabolites such as N-acetyl dopamine and N-acetyl tyramine, along with the measurement of free drug concentrations in the fly brain. These changes were observed following exposure to volatilized cocaine and methamphetamine. To follow the study objectives, brain samples were collected from flies at various time intervals ranging from 1 to 7 hours after exposure to the drugs. The LC-MS/MS was used to analysis of these targeted neurotransmitters and their metabolites. The study exhibits a compelling scientific background and holds significant interest within the field.

To enhance the quality of the manuscript, the following recommendations are suggested:

1.     In the title, ‘D. melanogaster’ can be updated with the full biological name ‘Drosophila melanogaster’.

2.     Lines 44-46. “In Drosophila……by psychostimulants.” The statement needs to be supported with a specific reference.

3.     Lines 99-102, The administration…..for vMETH. Hope these statements are based on some hypothetical assumption, not supported by any experimental data.   

4.     Lines 112 and 133, “close to 50% of the initial concentration was still present” and “50% was still present after seven hours”, how these statements are supported by the results. Please refer the figures 1A and 2A.

5.     In the figures, most of the parameters were not supported with specific units.

6.     Line 122, the doi of the reference (doi.org/10.3390/toxins14040278) has been replaced with a citation.

7.     Lines 128-131, 248-250, also from figures 1B-1D, after three hours following the administration of cocaine, observed a significant and distinct spike in the levels of dopamine, tyramine, and octopamine, which was not observed with methamphetamine. To further strengthen the validity of these findings, the authors can include additional supporting evidences in the discussion section.

8.     Line 314, “vMETH administrations does not vary significantly over seven hours”, to be improved.

Round 2

Reviewer 1 Report

The authors have done more than adequate in responding to the comments raised by me.